# “No Pain No Gain”: Evidence from a Parcel-Wise Brain Morphometry Study on the Volitional Quality of Elite Athletes

**DOI:** 10.3390/brainsci10070459

**Published:** 2020-07-17

**Authors:** Gaoxia Wei, Ruoguang Si, Youfa Li, Ying Yao, Lizhen Chen, Shu Zhang, Tao Huang, Liye Zou, Chunxiao Li, Stephane Perrey

**Affiliations:** 1CAS Key Laboratory of Mental Health, Institute of Psychology, Chinese Academy of Sciences, Beijing 100101, China; yaoy@psych.ac.cn (Y.Y.); chenlz@psych.ac.cn (L.C.); zhangs@psych.ac.cn (S.Z.); 2CAS Key Laboratory of Behavioral Science, Institute of Psychology, Chinese Academy of Sciences, Beijing 100101, China; 3Department of Psychology, University of Chinese Academy of Sciences, Beijing 100049, China; 4CUBRIC, Cardiff University, Maindy Road, Wales Cardiff CF24 4HQ, UK; SiR@Cardiff.ac.uk; 5Collaborative Innovation Center of Assessment toward Basic Education Quality, Beijing Normal University, Beijing 100875, China; liyoufa@bnu.edu.cn; 6Department of Physical Education, Shanghai Jiao Tong University, Shanghai 200030, China; taohuang@sjtu.edu.cn; 7Department of Psychology, Shenzhen University, Shenzhen 518061, China; liyezou123@gmail.com; 8School of Physical Education and Sports Science, South China Normal University, Guangzhou 510006, China; chunxiao.li@nie.edu.sg; 9EuroMov Digital Health in Motion, Univ Montpellier, IMT Mines Ales, 34090 Montpellier, France; stephane.perrey@umontpellier.fr

**Keywords:** volition, brain structure, sense of agency, sport, MRI

## Abstract

Volition is described as a psychological construct with great emphasis on the sense of agency. During volitional behavior, an individual always presents a volitional quality, an intrapersonal trait for dealing with adverse circumstances, which determines the individual’s persistence of action toward their intentions or goals. Elite athletes are a group of experts with superior volitional quality and, thereby, could be regarded as the natural subject pool to investigate this mental trait. The purpose of this study was to examine brain morphometric characteristics associated with volitional quality by using magnetic resonance imaging (MRI) and the Scale of Volitional Quality. We recruited 16 national-level athletes engaged in short track speed skating and 18 healthy controls matched with age and gender. A comparison of a parcel-wise brain anatomical characteristics of the healthy controls with those of the elite athletes revealed three regions with significantly increased cortical thickness in the athlete group. These regions included the left precuneus, the left inferior parietal lobe, and the right superior frontal lobe, which are the core brain regions involved in the sense of agency. The mean cortical thickness of the left inferior parietal lobe was significantly correlated with the independence of volitional quality (a mental trait that characterizes one’s intendency to control his/her own behavior and make decisions by applying internal standards and/or objective criteria). These findings suggest that sports training is an ideal model for better understanding the neural mechanisms of volitional behavior in the human brain.

## 1. Introduction

Volition or the will is a univocal concept with a long history in philosophy. From a psychological perspective, it has been considered as a psychological construct, being used to describe one’s endogenous mental act of forming, maintaining, and implementing an intention or goal [1,2]. This mental process or conscious experience is unique to human beings and associated with a sense of agency [3]. From the developmental perspective, volitional quality is an intrapersonal trait for dealing with adverse circumstances that determines an individual’s persistence of action upon intentions or goals [4,5]. This capability is of great significance for individual survival, development, and achievement.

Distinguished from concepts or terms such as grit [6], willpower [7], mental toughness [8,9], and resilience [10], volitional quality has a great emphasis on the sense of agency within four components (self-conscientiousness, independence, determination, and resilience) [5,11,12,13,14]. This theoretical model has been widely used in education, physical activity, and sports settings. For instance, junior school students who regularly participated in rock climbing showed greater levels of volitional quality than those without regular participation [15]. A study observed that a four-month outdoor survival training program significantly improved performances in independence, determination, and resilience among college students [16]. Even three-month-long professional sports training was found to enhance an individual’s volitional quality [17]; however, the brain structures related to volitional quality are largely unexplored.

With the advance of human brain imaging techniques, a growing body of evidence indicates that the brain structure can be reorganized through learning and life experiences. A magnetic resonance imaging (MRI) study that was performed on experts suggested that the brain volume of the posterior hippocampi in taxi drivers was significantly larger than those of age-matched controls, and the greater brain volume of the posterior hippocampi was positively associated with their driving experiences [18]. Meditators and Tai Chi Chuan masters also showed greater cortical thickness in multiple brain areas related to high-level cognitive processes as compared with the sedentary controls [19,20]. Additionally, some longitudinal studies provide direct evidence on use-dependent brain plasticity. Currently, a growing body of evidence has demonstrated that physical training could induce structural plasticity. Rogge, et al. [21] investigated the effect of a twelve-week whole-body training with minor metabolic demands on the brain structure, which showed that balance training could increase the cortical thickness in the visual and vestibular cortical regions. A recent study indicated that the alteration of the volume and cortical thickness of grey matter could be induced by three-week motor training in adults [22].

Elite athletes are a group of experts with extraordinary physical abilities and mental attributes that are developed over long-term sports training. Professional training may also have a pronounced effect on cortical organization, and some neurocognitive researchers have examined the brain plasticity in elite athletes. For instance, elite short track speed skating (STSS) players exhibited larger volumes in the right cerebellar hemisphere and vermian lobules VI-VII relative to the matched controls [23]. Our previous studies observed that elite diving players showed greater grey matter density in the thalamus and the left precentral gyrus [24] as well as greater cortical thickness in the left superior temporal sulcus, the right orbitofrontal cortex, and the right parahippocampal gyri, as compared to the controls [25]. Similarly, highly practiced golf players showed larger gray matter volumes in the frontoparietal network relative to controls without any golfing practice and less experience [26]. Another study on male adolescent elite footballers showed that increased training volumes improve the cortical area [27]. Overall, these findings suggest that sports training may reorganize cortical and subcortical structures.

Winter sports are sports that have great requirements for physical endurance and independent decision-making capabilities to navigate the snow and ice equipment. Of note, short track speed skate (STSS) athletes who engaged in winter sports often experienced many physical and mental setbacks, limb injuries, and performance slumps, providing a unique opportunity to investigate their volitional quality. A recent behavioral study involving 169 winter sports athletes (alpine skiers) by a retrospective design found that their grit and perfectionistic strivings were closely associated with increased engagements in practice hours, suggesting winter sports athletes might have outstanding volitional qualities [28]. Skaters tend to peak at their late teens and early adulthood, the age at which people also often experienced a dramatic cortical reorganization; therefore, alterations in cortical structures associated with volitional quality are expected in winter sports players. In this study, we aimed to compare elite athletes and control subjects through vertex-wise and parcel-wise analyses to detect multiple morphological differences. Both of two approaches are based on surface-based morphometry. The vertex-wise analysis is a method used to compute local morphological parameters based on each voxel, whereas the parcel-wise method is used to analyze large-scale cortical organization at the level of cortical parcellation [29]. The use of both vertex-wise and parcel-wise analyses can help researchers better understand the gray matter structures from different levels. In order to comprehensively understand mental trait-related anatomical correlates, the combination of two analysis approaches in one study is an ideal approach to completely measure the cortical morphometric changes behind the training-induced mental traits. Using MRI, we aimed to identify the cortical organization associated with volitional quality by comparing their anatomical differences between STSS elite athletes and control subjects. Specifically, we recruited STSS elite athletes from the Chinese national team who are world-class athletes and have experienced many international competitions as an expert group.

## 2. Methods

### 2.1. Ethics Statement

This study was approved by the institutional review board of the Institute of Psychology, Chinese Academy of Sciences (Approval date: 20180316). It was performed following the ethical standards laid down in the 1964 Declaration of Helsinki. The written informed consent forms were obtained from all participants and their parents/guardians for those participants aged under 18 years.

### 2.2. Participants

The participants were 34 right-handed young adults, including 16 STSS athletes from the Chinese national team (age: 18.3 ± 1.5; 7 males) with extensive sports training experiences (8.6 ± 1.9 years) and 18 college students as controls (age: 19.2 ± 1.2; 9 males) matched for sex and age. The participants in the control group were recruited from a local university and had no regular exercise or sports training experiences. All participants were right-handed and healthy. They did not have any history of substance dependence, e.g., alcohol or nicotine. Before the experiment, participants completed a screening form to ensure they did not have a history of hearing or vision problems, physical injury, seizures, metal implants, head trauma with loss of consciousness, or pregnancy.

### 2.3. Measures

#### 2.3.1. Volitional Quality

Participants’ volitional qualities were measured by the 36-item BTL-L-YZ 2.0 Scale of Volitional Quality [11,30]. This scale measures four dimensions of volitional quality, including self-consciousness (12 items), independence (8 items), determination (8 items), and resilience (8 items). The factorial validity, concurrent validity, internal reliability, and test-retest reliability of the scale were supported [11]. A 7-point Likert scale that ranged from 1 (strongly disagree) to 7 (strongly agree) was used for the responses.

#### 2.3.2. Scanning Protocol

An MRI was performed on a 3-tesla scanner (Discovery MR750, GE, USA) with a 12-channel head matrix coil. All high-resolution anatomical images were obtained using the spoiled gradient recalled acquisition in a steady-state (SPGR) sequence with the following scan parameters: echo time (TE) = 2992 ms, repetition time (TR) = 6896 ms, flip angle (FA) = 8°, slice thickness = 1.0 mm, and Field of View (FOV) = 256 mm × 256 mm. The imaging data included 176 sagittal slices. During scanning, all participants reclined in a supine position on the bed of the scanner and were asked to lie still during the imaging procedure. A foam head holder and padding were placed around their head. Moreover, headphones were provided to block background noise.

#### 2.3.3. Image Processing

After the image acquisitions, all images were visually checked for major artifacts, including head motions, brain lesions, and dissection, before further processing. Brain reconstructions from the structural images were conducted with the VolBrain system [31] (http://volbrain.upv.es/) and FreeSurfer (version 6.0), which were integrated into the pipeline of the Connectome Computation System (CCS, [32]). All individual images went through the same pipeline, including the following steps. First, a nonlocal mean-filtering operation was used to remove the spatial noise [33,34]. Second, the MR-inhomogeneity-induced image intensity variance was corrected [35]. Third, the brain images were extracted, and images containing nonbrain tissues were removed using a hybrid approach [36]. Fourth, the brain tissues were segmented into the cerebrospinal fluid (CSF), white matter (WM), and deep gray matter (GM), and the two hemispheres and subcortical structures were disconnected [35]. Fifth, the GM-WM boundary was tessellated with a triangular mesh and smoothed by a mesh deformation [35]. Sixth, the topographical defects were corrected on the surface, and the individual surface mesh was inflated into a sphere [37,38]. Finally, the individual surface (volume) was normalized by estimating the deformation between the individual brain volume and the common spherical coordinate system [39].

#### 2.3.4. Parcel-Wise Cortical Thickness Computation

It was computed by the *recon-all* command implemented in FreeSurfer (version 6.0). The local cortical thickness was measured by averaging the shortest distance from a vertex on the white matter surface to the pial surface and the shortest distance from a point on the pial surface to the white matter surface [40]. There are 163,842 vertices in each hemisphere, and there are 327,684 total vertices in the whole brain. This measurement has a good test-retest reliability across field strengths, scanner upgrade, and scanner manufacturers [41]. Individual cortical thickness maps were registered to the *fsaverage* and smoothed to enhance the interindividual correspondence in the anatomical structure using a Gaussian filter of 10-mm full width at half-maxima (FWHM) [42].

The Desikan-Killiany atlas [43,44] was employed to parcel the cortical surface into 34 regions of interest (ROIs) in each hemisphere to explore the large-scale structures. The mean brain morphometric parameter in cortical thickness in each parcel was then calculated for each participant.

### 2.4. Statistical Analysis

Vertex-wise group differences in cortical thickness were examined. For each vertex, a general linear model (GLM) was employed with the gender, age, education, and Intracranial Volume (ICV) entered as covariates. Between-group comparisons were then computed, and a cluster-level correction for multiple comparisons was conducted using the random field theory with vertex *p* < 0.001 and cluster *p* < 0.05. These analyses were conducted in FreeSurfer 6.0.

Parcel-wise group differences in cortical thickness were computed using analysis of covariance (ANCOVA). Gender, age, education, and ICV were included as covariates. The adjusted *p*-threshold for the group difference was set to 0.00073 (i.e., 0.05/(2*34)) using Bonferroni correction for a multiple comparison of 68 ROIs in total.

Multivariate analysis of covariance (MANCOVA) was conducted to test the group difference in four dimensions of volitional qualities, with gender, age, and education as covariates. The *p*-value was set to 0.0125 (i.e., 0.05/4) using Bonferroni correction for multiple comparisons of four dimensions in total.

To examine the association between cortical thickness and volitional quality in each group, two separate partial correlation analyses were performed, with gender, age, education, and ICV as covariates. We also performed a partial correlation between the differed cortical thickness in the brain regions and training experiences in the athlete group, with the age and sex as covariates. The *p*-value was set to 0.05/*n* using Bonferroni correction for multiple comparisons of the brain regions with significant difference. These analyses were conducted with SPSS Statistics 22.0 (IBM Corp., Armonk, NY, USA).

## 3. Results

### 3.1. Demographic Data

There was no group difference in age (*t*(32) = −1.994, *p* = 0.055), gender (*χ^2^* = −0.133, *p* = 0.716), body mass index (BMI) (t = −0.850, *p* = 0.401), and ICV volumes (*t*(32) = −0.706, *p* = 0.485); however, the control group had finished more years of education than the athlete group (*t*(32) = −9.962, *p* = 0.000) (Table 1).

### 3.2. Group Difference in Volitional Qualities

The athlete group showed significantly higher scores in all dimensions, including self-conscientiousness ((*F*(1,29) = 12.086, *p* = 0.002), independence (*F*(1,29) = 28.386, *p* < 0.001), determination (*F*(1,29) = 7.850, *p* < 0.001), and resilience (*F*(1,29) = 7.842, *p* < 0.001), relative to the control group (Table 2).

### 3.3. Group Difference in Cortical Thickness

The results of the vertex-wise analysis did not show any significant group differences after a multiple comparison correction. The results of the parcel-wise analysis showed that the athlete group had significantly greater cortical thickness in the left precuneus (*F*(1,29) = 22.105, *p* = 0.00006), the left inferior parietal lobe (*F*(1,29) = 16.046, *p* = 0.00041), and the right superior frontal lobe (*F*(1,29) = 23.971, *p* = 0.00004) than the control group (Figure 1 and Figure 2).

### 3.4. Correlation Analyses

Regarding three brain regions with significant differences of cortical thickness, the adjusted *p*-value was set to 0.05/3 (0.017). As Figure 3 showed, the score of independence was significantly correlated with the cortical thickness in the left inferior parietal lobe (*r* = 0.701, *p* = 0.011). The other three dimensions showed no association with cortical thickness (*r* = −0.335 to 0.609, *ps* > 0.017). However, we did not observe any significant correlation between years of training and the score of any volitional qualities in the athlete group (*r* = −0.001. to 0.372, *ps* > 0.05).

## 4. Discussion

To the best of our knowledge, this is the first brain imaging study to identify cortical architecture associated with volitional qualities. Our study found that STSS athletes showed better volitional qualities than the control group. Regarding the differences in brain morphometry, the athletes group had greater cortical thickness in the left precuneus, the left inferior parietal lobe, and the right superior frontal lobe. The greater cortical thickness in the left inferior parietal lobe was significantly associated with dimension independence of the volitional qualities.

As hypothesized, the behavioral results showed that athletes had better volitional qualities relative to the controls. In addition to the total score of the volitional qualities, all dimensions, including consciousness, independence, determination, and resilience, were significantly higher in the athlete group compared to the control group. This result is consistent with a previous study, which also revealed that STSS athletes scored higher in goal clarity, persistence, determination, and confidence compared to the controls [45]. Similarly, a study on national-level professional basketball players observed that they showed better performances in consciousness and independence than players in other levels [46]. Nonetheless, some studies on other antagonistic sports observed that professional players had significantly better performances in the total scores of volitional qualities but showed different advantages in specific dimensions. For instance, a study on boxers showed a significant correlation between competition scores and resilience in volitional qualities [47]. Moreover, another study comparing the effect of training experience on volitional qualities showed that beach volleyball athletes with above 15 years of training experience had better continence and resilience than those under 15 years of training experience [48]. All these findings indicate that sports training selectively improves the components of volitional qualities. STSS is a sport that requires the adjustment of pacing and tactical positioning [49]. It was found that the optimal pacing strategy varies among STSS projects of different distances. A 500-m race needs a fast start, whereas a 1500-m race has a greater emphasis of physical exertion in the last five laps [50,51]. So, STSS athletes need to make decisions on their own regarding how to distribute the energy and what moment to invest their energy during the race bout [52]. Moreover, they have to keep clear goals during the competition and adhere to a chosen strategy throughout the whole bout. Therefore, in this study, we observed that long-term professional training might improve their mental characteristics of volitional qualities.

The basis of volitional qualities is closely related to the sense of agency [3]. A sense of agency is also called the sense of control, i.e., a subjective awareness of initiating, executing, and controlling one’s own volitional actions in the world, as well as the experience of oneself as the agent of one’s own motor acts [53]. Therefore, it is accepted that our experiences of volitional behaviors include a vivid sense of agency [3,54,55]. Of note, dimension independence in volitional qualities is a mental construct that is closely associated with a sense of agency, which is a mental trait that characterizes one’s tendency to control his/her own behavior and making decisions by applying internal standards and/or objective criteria [56]. An individual having a good sense of agency is capable of controlling their behaviors, which is a prerequisite for volitional qualities and independence. On the other side, impairments in the sense of agency have been reported in neurological and psychiatric disorders [57], indicating that these patients find it difficult to control their own behaviors and present poor volitional qualities. The athletes recruited in this study are professional athletes engaged in their career for at least ten years and have rich experiences in dealing with difficulties in important national and international competitions. It is believed that a sense of agency is strongest when there is a strong motivation to act with a clear goal [58]. Although the mechanism underlying the association of sports training and the sense of agency remains largely unclear, it has been reported that the sense of agency was improved by listening to music in one’s daily life [59]. Hence, it is likely that long-term sports training under adverse circumstances likely reshapes elite athletes’ sense of agency, which further enhances their volitional qualities.

Intriguingly, all of these brain regions (the precuneus, the inferior parietal lobule, and the superior frontal cortex) are consistently linked to the function of the sense of agency [60]. Precuneus is a posteromedial portion of the parietal lobe responsible for processing self-relevant information. A task-based fMRI study observed that the bilateral anterior precuneus was activated when participants were exposed to psychological trait words describing their own attributes [61]. Apart from the explicit external information, the precuneus is also involved in implicit internal self-related processes, especially in making intentional behaviors [62]. During this process, the precuneus contributes to the sense of agency [53,60]. A recent study investigated the focal lesion areas among 50 patients with a disrupted sense of agency and found a persuasive causal relationship between the intactness of the functional networks related to the precuneus and a sense of agency [55]; therefore, it is speculated that individuals with good performances of independence have a better sense of agency, which is tightly linked to the greater cortical thickness of the precuneus. In addition to the precuneus, the left inferior parietal lobule (IPL) is another anatomical difference related to the sense of agency. A study observed that the IPL was less-activated when the right-handed subjects felt that the behavior was performed by themselves compared to when they felt that the behavior was not performed by themselves [63]. The frontal and prefrontal lobes play a crucial role in the planning and initiation of voluntary action [64]. Notably, parts of the superior frontal gyrus are involved in inhibitory control. For instance, the inhibitory control was slowed in patients with right superior medial frontal damage [65]. Another study also found that the right superior frontal cortical activations to conflict anticipation are related to impulse control in healthy participants [66]. The inhibitory control is a cognitive function also associated with the sense of agency, since a fluency of action selection contributes heavily to the sense of control [67]; therefore, it indicates that the greater cortical thickness in the right superior frontal gyrus might also be associated with an improved sense of agency. Although it is unclear what the underlying mechanism is that causes different cortical thicknesses between the athlete group and the control group, an increased cortical thickness is likely associated with a complex change in the microstructure, e.g., the increases in dendritic arborization, axonal elongation and thickening, synaptogenesis, and glial proliferation [68,69]. Numerous animal models and human studies have consistently reported that training selectively improves neurogenesis and induce changes in the cortical volume [70]. Therefore, our results of greater cortical thicknesses in these brain regions reflect the training-induced changes in the cellular and neuronal levels.

Several limitations should be acknowledged in this study. First, we could not conclude the causal relationship between volitional quality and parcel-wised cortical thickness. Future investigation on a less-skillful amateur group is needed to elucidate whether the difference is induced by training or nurture. Secondly, the relationship between the cortical thicknesses and the years of training was not evident. Instead of years of training, training volume and/or intensity might have a closer association with the changes in the cortical thickness [19,25]; therefore, multiple indices to measure training experiences such as the training load (intensity x volume) should be used in future research to gain a better understanding about the issue. Thirdly, the cross-sectional study design could not completely exclude the confounding effects of nature or nurture on the brain structures in the two groups. These confounding variables include individual differences such as lifestyle, nutrition, and personality. In this study, the difference in education between the two groups might be a potential factor that influences the current results. In future studies, a longitudinal study is needed to rule out the effects of individual differences.

## 5. Conclusions

Although some limitations existed in this study, it still identified the cortical architecture associated with volitional qualities. Professional athletes exhibited excellent volitional qualities, as well as thicker cortexes in the left precuneus, the left inferior parietal lobule, and the right superior frontal gyrus. Better performance in the dimension independence of the volitional qualities was correlated with a greater cortical thickness in the left inferior parietal lobule. These findings suggest that sports training is an ideal model for better understanding the neural mechanisms of volitional behaviors in the human brain.

## Figures and Tables

**Figure 1 brainsci-10-00459-f001:**
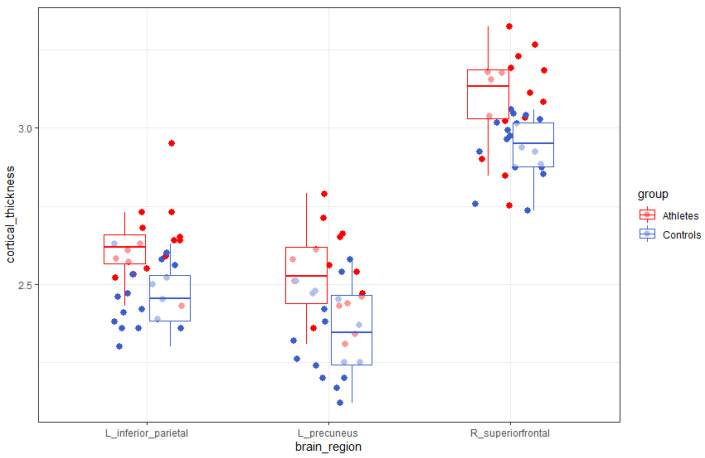
Parcel-wise difference of cortical thickness in the left precuneus, the left inferior parietal lobe, and the right superior frontal lobe between the athlete group and the control group. Data are presented by dot plots, whiskers, and box plots (median and error bars representing the first and third quartiles, respectively).

**Figure 2 brainsci-10-00459-f002:**
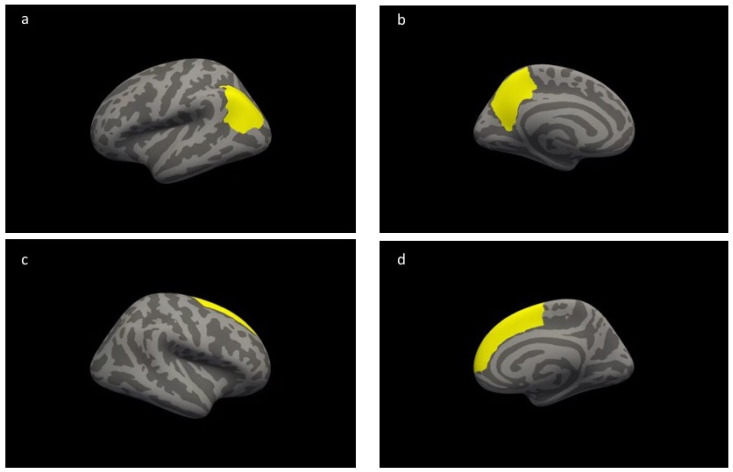
Compared with the control group, the athlete group showed greater cortical thickness in the colored brain regions based on the parcel-wise analysis: (**a**) indicates the left inferior parietal lobe, (**b**) indicates the left precuneus, and (**c**,**d**) the lateral view of the inferior view of the right superior frontal lobe.

**Figure 3 brainsci-10-00459-f003:**
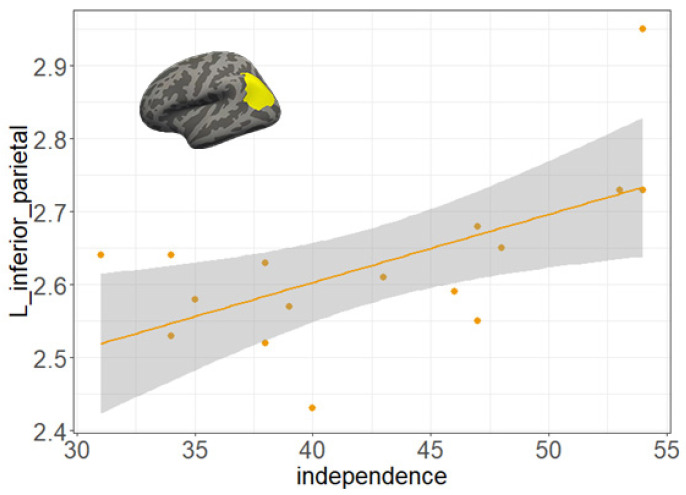
Scatter plots indicating the significant correlations between the independence scores of the volitional qualities and cortical thickness of the left inferior parietal lobule.

**Table 1 brainsci-10-00459-t001:** Demographic data of athletes and controls.

	Age	Gender	Education (Years)	Body Mass Index (BMI)	Years of Training	Intracranial Volume (ICV) (cm^3^)
Athletes (*n* = 19)	18.3 ± 1.5	10 M	9.8 ± 1.8	21.2 ± 1.1	8.6 ± 1.9	1178.06 ± 85.20
Controls (*n* = 19)	19.2 ± 1.2	9 M	13.7 ± 1.0	22.0 ± 3.9	—	1210.68 ± 87.10

**Table 2 brainsci-10-00459-t002:** Group differences in dimensions of volitional quality.

Dimensions	Athletes (M ± SD)	Controls (M ± SD)	F	*p*
Self-Conscientiousness	58.38 ± 7.99	45.72 ± 4.35	28.386	0.000
Independence	42.56 ± 7.53	27.50 ± 2.23	12.086	0.002
Determination	35.56 ± 4.75	27.11 ± 3.25	7.850	0.009
Resilience	32.94 ± 4.40	23.11 ± 2.59	7.842	0.009

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
