# Peer review of "“No Pain No Gain”: Evidence from a Parcel-Wise Brain Morphometry Study on the Volitional Quality of Elite Athletes"

_brainsci, 2020, doi:10.3390/brainsci10070459_

Round 1
Reviewer 1 Report
In the present manuscript, Wei and colleagues sought to relate differences in brain structure and volitional quality in elite athletes and healthy controls. While the topic and hypotheses are generally interesting, a number of revisions are recommended.
General
- Extensive editing of the English language is required.
- The title indicates this is a study of elite athletes, but healthy controls (who also have 'volitional quality') are also included. A revised titled is suggested.
Introduction
- In the second sentence, the authors claim that there is no 'clear concept of volition', which calls into question the basic premise of, and measurements used in, this study. This contradicts the clear definitions in the 2nd paragraph.
- The authors are advised to avoid the term 'neural correlates'. The authors lay out a rationale whereby training could cause changes in brain structure ('bottom-up') and differences in brain structure could have implications for behavior ('top-down'). Therefore more specific language is warranted when discussing the precise relationships between the dependent and independent variables in this study. Yes, the relationship observed in this study are correlational, but the rationale for the measures and the speculation to their relevance has 'direction'.
- Additionally, it is not clear to this reader the degree to which 'cortical thickness' (as estimated from FS parcellations) represents anything 'neural'. If the term remains, a proper definition in the context of these measures should be provided.
- The hypotheses make this seem like a fishing expedition.
Methods
- Please clarify that the list of steps in Image Processing is an explanation for the FS pipeline and not performed in preparation for or in addition to the FS pipeline.
- Why do both parcel and vertex based analyses? Again, without clear hypotheses this seems like a fishing expedition.
- Considering the readership of this journal, please specify how many vertices are produced by reconall so the general reader better understands how many GLMs were performed.
- One does not 'set' a p-value. I think you mean that you set an alpha level (i.e. p-threshold) based on these Bonferonni corrections.
- Regarding the MANCOVAs: Please provide a rationale for correcting for ICV when comparing volitional quality between groups?
- Why were partial correlations used for the brain/behavior tests and not regressions, which would be more similar to the ANOVAs used to this point? Though similar, the handling of variance between PCC and regression is different. Regardless, correction for multiple comparisons should be the same as the ANCOVAs, but none is mentioned.
Results
- The hemisphere comparisons between groups was not expected. Are left/right hemispheres ROIs in the D-K atlas? If so, please specify in Image Processing. If not, please specify this comparison in Statistical Analysis. Or maybe the authors just added up all ROI thicknesses by hemisphere manually?
- Please indicate whether p-values are corrected or uncorrected.
- I suspect they are corrected, but Figure 2 is not at al convincing. It is going to be very difficult to convince a reader that those relationships (the pearson correlation coefficients) have an uncorrected p-value of <.00073.
- Are the grey bars in Fig. 2 95% confidence intervals?
Discussion
- There is a disconnect between what was measured and what is discussed as the authors abandon the factor structure of their 'volitional quality' measure. Throughout this manuscript the reference to 'sense of agency' has been confusing. Per Haggard (as cited), the reader is encouraged to take this to mean 'a feeling of making something happen'. From a neuroscience perspective, this seems like an extension of Damasio's work and more generally as a term representing sensorimotor integration. This is interesting in the athletic context! However, the authors re-define this in paragraph 3 as 'sense of control' and much of the ensuing discussion is about 'cognitive control' or 'inhibitory control'. These were not measured. In paragraph 4 the authors state that the relationship between independence and sense of agency is entirely speculative. Major revisions to the discussion are strongly recommended.
- The authors measure brain structure, but most of this discussion (paragraphs 3-5) attempts to ascribe functional meaning to differences in thickness. This makes the discussion is highly speculative and reads as a combination of reverse inference and modern-day phrenology.
- Paragraph 3: The opening two sentences do not accurately represent the results. (right SFG was not related to vol. quality?) What is 'first-person view'?
- Paragraph 4: The SFG is large and suggesting that the entire region plays a role in any single function is misleading ("notably, the right SFG has a specific involvement in inhibitory control").
- Paragraph 5: The authors suggest that precuneus thickness holds implications for participant 'mental state' (in the scanner...and beyond?) because of its inclusion as a node in the DMN, which is an intrinsic functional connectivity network.
- This reviewer is not aware of solid scientific basis for this claim.
- Though this reviewer is not clear on what 'self-generated thoughts' are, it seems like a deviation from 'sense of agency' which would represent the junction between interoception and behavior.
- What is meant by 'recruiting' the DMN? This seems anti-thetical to the very basis of the DMN. When recruiting task-oriented networks, the DMN would be more silent. Again, what any of this means in the context of cortical thickness is unclear.
- The authors consistently attempt to attribute one function to one region, which is not how we understand the brain to function. (e.g. precuneus = "processing self-relevant information"; right SFG = "inhibitory control", precuneus = "self-generated thoughts".
- Why were the differences in thickness (and relationship between thickness and behavior) unilateral and not bilateral? Please discuss.
- How would longitudinal data support causation? Sure, two factors (structure & behavior) can change together over time, but there are many more gaps in logic to reason out the direction of the effect between structure<-->behavior.
Conclusion
- See previous points about DMN.
Reviewer 2 Report
brainsci-844485-peer-review-v1
No Pain No Gain”: Evidence from A Parcel-Wise Brain Morphometry Study on Volitional Quality of Elite Athletes
The authors used T1-MRI to investigate alterations in the brains grey matter volume between 16 athletes (short track speed skating) on a national competitive level and 18 controls. They used parcel wise brain anatomical characteristics and detected three areas with increased grey matter volume in athletes: left precuneus, left inferior parietal lobe, and right superior frontal lobe.
The authors investigate an important topic here. Other investigations on volitional training, VBM studies on expertise in different aspects of sports and expertise, are however not mentioned here.
Abstract:
Change “magnet resonance imaging (MRI)“ to “magnetic resonance imaging (MRI)“
Writing unusual („volitional quality“ two times in one sentence)
The construct „volitional quality „ not explained-what is that and how was it evaluated here? The same holds to be rue for the parameter “independence”.
Abbreviation “DMN” not defined here.
Introduction:
“However, neural correlates of volitional capacity are largely unexplored.“ I think the authors did not look for papers describing these. Use the terms „volition“, „training“ „expertise“, „cognitive expertise“ and combine with brain imaging methods such as „MRI, TMS, VBM, ALE-metaanalysis“. They did even not cite „Jäncke L, Koeneke S, Hoppe A, Rominger C, Hänggi J. The architecture of the golfer's brain. PLoS One. 2009;4(3):e4785.“ Instead they cite studies on taxi drivers and tai chi meditation techniques.
It is not clear why STSS was selected here.
Methods:
Characterization of participants is not sufficient. This should include mood, alcohol and nicotine, and testing of some cognitive abilities. How much exercise did the HCs have? It could be a very selective difference in other items instead oft he expertise in sports and volition which drives the differences between groups.
Did the authors use templates here? What template might be the best then for that age group?
Use of 34 parcels per hemisphere; multiple comparison correction therefore on 68 regions seems reasonable. But spatial resolution is quite poor with this procedure.
Results:
The athletes had greater cortical thickness in general than controls.
In addition cortical thickness was increased inn 3 brain regions. The Figures are appropriate but a Figure on location of that parcel might be added here.
Correlation analyses: please also add a small brain figure on the ROI-location.
Discussion:
Overall the discussion is quite strong.
There are some issues which might be mentioned:
The authors did not control for other factors which have an impact on the brains grey matter such as body mass index, nicotine, alcohol. Education seems to differ between groups; please discuss possible impact on results.
The athletes had greater cortical thickness than controls. Both were about 18-19 years old. In that age the amount of sports in spare time negatively associates with the brains gray matter volume since the brain is in permanent reshaping to be more effective. Therefore, those who are more effective have usually smaller volume in ROIs.This is contrary to findings in elderly where movement is protective for age decline related gray matter decrease (see for instance Eyme et al., 2019). Is that a problem of methodology (cortical thickness or GMV)?
Lack of stat. power should be discussed. I think the low statistical power is the reason for the poor spatial resolution (68 parcels) the authors selected for testing.
Minor:
Formal: problems with spacing all over the manuscript.

Reviewer 3 Report
I received for expertise the article entitled “No pain No gain”: Evidence from a parcel-Wise brain morphometry study on volitional quality of elite athletes. The study aims to examine brain morphometric characteristics associated with volitional quality by using magnetic resonance imaging (MRI). After expertise, my remarks are as follows:
- Overall, the authors should read their paper carefully to avoid mistakes. for example:
- Abstract - line 4: “magnetic” instead of “magnet”.
- Discussion - paragraph 1 - line 6: delete the repeated word “in”
- Discussion - paragraph 5 - line 1: “frontal” instead of “fontal”
- Abstract
- Avoid the abbreviation DMN
- Introduction
- Paragraph 2 - line 6: Put the reference [16] at the end of the sentence. “Students [16]”
- The authors should complement the introduction with the effects of physical exercise on cortical thickness in young people. See the works: DOI: 10.1016/j.neuroimage.2018.06.065 - DOI: 10.1016/j.neuroimage.2016.10.033 - DOI: 10.1055/s-0042-124510
- Methods
- Participants: How inactivity was measured in the control group.
- Measures: it would be interesting for the authors to put the Scale of Athlete’s Volition questionnaire as an appendix or a supplementary file.
- Results
- In Table 2, the p-values (0.009) show that there is a significant difference between the two groups on determination and resilience. I don’t understand why the authors say: “while no significant differences were observed in the other two dimensions”. However, the authors specify that the p-value is 0.0125.
- The figure is not readable. In addition, since there are two graphs on the figure, authors should name figures a and b as they did for figure 2.
- Discussion
- Paragraph 4 - line 2: To standardize the following reference (Haggard, 2017)
- In my opinion, the discussion lacks the mechanisms to explain the difference in cortical thickness between the two groups. This seems to me to be essential.
Round 2
Reviewer 2 Report
The manuscript has much improved and all my concerns have been adressed now.
Reviewer 3 Report
Good job